# Vaginal Microbiome Dynamics of Cows in Different Parities

**DOI:** 10.3390/ani13182880

**Published:** 2023-09-10

**Authors:** Jiale Ni, Jie Wang, Kaisen Zhao, Yang Chen, Siqi Xia, Songjia Lai

**Affiliations:** College of Animal Science and Technology, Sichuan Agricultural University, Chengdu 611130, China; njl2077@163.com (J.N.); wjie68@163.com (J.W.); zhaofl0303@163.com (K.Z.); chenyi154121@163.com (Y.C.); xiasiqi2020@163.com (S.X.)

**Keywords:** high-throughput 16S rRNA sequencing, cow, vaginal microorganisms, different parities

## Abstract

**Simple Summary:**

At present, research on the vaginal microbiome of dairy cows is mostly cross-sectional, and there is still room for research on the relationship between the vaginal microbiome and cow reproductive age. In this study, high-throughput 16S rRNA sequencing technology was used to explore vaginal microbiota differences in dairy cows of different parities and thus gain an in-depth understanding of cow reproductive physiology. Studies have found that vaginal microorganisms in cows change significantly before and after the first calving, and the importance of the first production of cows to the establishment and reproduction of subsequent vaginal microorganisms may be more important than previously thought. There was no significant change in the vaginal microbiota of calves in subsequent production, and the number of vaginal microorganisms gradually decreased with the increase in parity, but the vaginal microbiota tended to be stable. The specific mechanism of vaginal microbiome changes in dairy cows needs further study.

**Abstract:**

At present, there is still room for research on the relationship between the vaginal microbiome and the reproductive health of dairy cows. In this study, high-throughput 16S rRNA sequencing technology was used to explore the differences of bacterial communities of dairy cows of different births, gain a deeper understanding of cow reproductive physiology, and maintain cow health. With the increase in parity, the number of vaginal flora decreased from 3511 to 469, but the number of species increased significantly, and Chao1 increased from 1226.41 ± 345.40 to 1467.76 ± 269.76. There was a significant difference in the number of vaginal microbiome functions between uncounted cows and calving cows. There was no significant difference in microbial diversity in calves. The relative abundance variation of vaginal microbiota in high-parity cows is less than that in low-parity cows. The amino acid metabolism of calves increased, the endocrine function of high-parity cows was enhanced, and the function of the vaginal microbiome increased after the first delivery, which gradually decreased with the increase in parity. This study also found that *Methanobacteria* and *Caviibacter* may be involved in amino acid metabolism and endocrine function, and they may play a key role in cow reproduction. This study provides an important theoretical basis for studying changes in vaginal microorganisms in dairy cows, improves the understanding of reproductive health and production performance, and provides a scientific basis for improving the reproductive management of dairy cows.

## 1. Introduction

Mammalian organs and systems, especially reproductive and digestive tracts, possess a complex and dynamic microbiome. Reproductive tract microbes have a profound impact on endometrial health, the stability of the internal environment and fertility [1,2,3]. As an illustration, in women, the abundant presence of lactobacilli in the vaginal environment acts as a shield for the mucosal lining of the vagina through a competitive effect that impedes the attachment of pathogens [3,4]. The reproductive tract of healthy cows contains thick-walled bacteria, including *lactobacillus*. These lactobacilli, as probiotics that can be injected directly into the vaginal cavity, might play a pivotal role in controlling postpartum uterine diseases [1,5]. Furthermore, the administration of Saké *Lactobacillus* and *Lactococcus lactis* subspecies *cremoris* intravaginally during calving might potentially decrease the prevalence of postpartum endometritis [6,7]; likewise, the inclusion of lactobacilli helps regulate the pH balance in the vaginal area while supporting reproductive health by suppressing harmful pathogens, improving egg quality, and promoting proper corpus luteum function [8,9,10]. The outcomes of 16S rRNA sequencing on samples collected from the vagina and uterus demonstrated that a significant proportion of bacteria known to cause infections are commonly linked with inflammation of the uterus, endometritis, and infertility (e.g., *Clostridium* spp. and *Corynebacterium* spp.) [11,12].Research suggests that the microbiome of the reproductive tract in the postnatal period might have a significant impact on future reproductive outcomes [13,14].

Recent research into the vaginal microbiome of cows has revealed a complex and dynamic relationship between variations in bacterial community composition and their impact on bovine health. Fungal vaginitis can cause significant harm to the dairy industry, negatively affecting reproductive efficiency, milk production, artificial insemination schedules, calving rates, and increasing costs associated with treating infertile animals [15,16], The complexities surrounding the establishment of microbial communities within the bovine reproductive system have not yet been fully understood. Ongoing efforts are needed to clarify the intricate relationships among various factors and their impact on bovine reproductive health outcomes [17].

However, most current research on the vaginal microbiome of dairy cows is cross-sectional, and changes in the vaginal microbiome are a dynamic process, so comparisons of the vaginal microbiome in cows of different parities contribute to a better understanding of vaginal microbiome trends and their relationship to reproductive health [17].

The aim of our study was to compare the vaginal microbes of cows of different parities to study the differences in the vaginal bacteria community of cows in different cycles, to gain insight into the reproductive physiology of cows and to maintain herd health.

## 2. Materials and Methods

### 2.1. Animal and Sample Collection

Vaginal samples were collected from 224 healthy Simmental cows selected from the Yangping Cattle Farm in Sichuan, China. “G” represented the size of the parity for each group: G0 (n = 61), G1 (n = 60), G2 (n = 30), G3 (n = 28), G4 (n = 18), G5 (n = 14), and G6 (n = 13). G0 cows were all 9–12 months old, while G1–G6 were sampled 60–90 days after calving to ensure that cows were in the same physiological cycle. Dry and clean, spacious and comfortable farrowing rooms or farrowing houses were arranged during this period to support recovery and milk production. In this sense, the free movement of cows in the enclosure was ensured. Cows with health problems did not participate in sample collection. Cows were artificially inseminated during the second estrus period after giving birth, generally 50–90 days postpartum, and the gestation period was 280 days. All cows were fed with tie pens and fed TMR feed. To ensure cleanliness, we sanitized the cow’s vulva using alcohol and a cotton ball. Then, we collected the sample by inserting a sterile, unopened vaginal swab into the vagina, transfering it into an unopened tube and wrapping it tightly with aluminum foil. We kept the samples on ice and transported them to the laboratory on the same day. The samples were stored at −80 °C until DNA extraction. All protocols involving animals have been approved by the Sichuan Provincial Research Committee on Animal Care and Use Biology (DKY-B2019302083), China.

### 2.2. DNA Extraction

According to the manufacturer’s instructions, DNA extraction was carried out using the TIANamp Bacterial DNA Kit book and stored at −80 °C until subsequent analysis.

### 2.3. Polymerase Chain Reaction Amplification and Sequencing Library Construction

Using the diluted genomic DNA as a template, PCR was performed using specific primers with barcodes, Phusion^®^ High-Fidelity PCR Master Mix with GC Buffer from New England Biolabs (Ipswich, MA, USA) and high-efficiency, high-fidelity enzymes selected according to the amplification region to ensure amplification efficiency and accuracy. PCR products were detected by agarose gel electrophoresis at 2% concentration, and equal amounts were mixed according to PCR product concentrations. After thorough mixing, PCR products were detected again using agarose gel electrophoresis at 2% concentration, and the target bands were recovered using the gel recovery kit provided by Qiagen. The library was constructed using the NEBNext^®^ Ultra™ IIDNA Library Prep Kit (Cat No. E7645, New England Biolabs), and the constructed library was quantified by Qubit and Q-PCR; after the library was qualified, it was sequenced using Illumina NovaSeq6000 (San Diego, CA, USA).

### 2.4. Taxonomic Classification and Statistical Analysis

This study replaced OTUs with ASVs to improve the accuracy, comprehensiveness, and reproducibility of marker gene data analysis. Accordingly, data from each group were pooled and plotted into Venn diagrams based on the observed ASVs to reflect the similarity and number of overlaps in each ASV group.

The samples were split from the downstream data based on barcode sequences and PCR amplification primer sequences, and the reads were spliced using FLASH (V1.2.11, http://ccb.jhu.edu/software/FLASH/, accessed on 11 October 2022) software to obtain Raw Tags. Finally, the Clean Tags were compared with the database using Vsearch (Version 2.15.0) software to detect chimeras and remove them to obtain the final validated data, i.e., Effective Tags. The final ASVs (Amplicon Sequence Variants) and feature tables were obtained by noise reduction using the DADA2 module and filtering out the sequences with abundance less than 5. Subsequently, the taxonomic sklearn module in the QIIME2 software (Version QIIME2-202006) was used to compare the obtained ASV with the database, and the species information of ASV was taxonomically assigned.

Alpha diversity indexes, i.e., species richness, Shannon [18], Simpson [19], Chao1 [20], dominance and Pielou [21] indices, were calculated in QIIME2 software [22]. The level of the diversity index value can reflect the complexity of the microbial community contained in the sample. In addition, the significant difference test results of the alpha diversity index between the groups can quickly find the groups with significantly increased or decreased species diversity, so as to combine biological treatment for further analysis.

Unifrac distances were calculated using QIIME2 software, and R software (Version 3.5.3) was used to plot PCoA. Among them, PCoA were used with the ade4 and ggplot2 packages in R software. LEfSe was used to complete the analysis of species with significant differences between groups. Among them, LEfSe analysis is completed by LEfSe software (Version 1.0), and the default setting is 4 for LDA score. R software was used to test the difference between the two comparison groups at six taxonomic levels of phylum, class, order, family, genus and species, and the *p*-values were screened out. The species with *p*-values less than 0.05 were screened out as distinct species between groups, while the T test also used R software to realize the significant differences of species at various taxonomic levels.

According to the results of ASVs obtained by noise reduction and research needs, the common and unique ASVs between different samples (groups) are analyzed, and when the number of samples (groups) is less than or equal to 5, it is drawn as a Venn graph, and when the number of samples (groups) is greater than 5, the petal diagram will be displayed. Each circle in the figure represents a (group) sample, the number of circles and the overlapping part of the circle represent the number of ASVs in common between the samples (groups), and the number without overlap represents the number of unique ASVs of the sample (group).

The gene function profile of their common ancestor was inferred using PYRCRUST (1.0.0) implemented in QIIME2. The Greengene database was used to construct the gene function prediction profile of the whole spectrum of archaea and bacterial domains, and the composition of the sequenced microbiota was mapped to the database to predict the metabolic function of the microbiota.

## 3. Results

### 3.1. Number of Sequences

A total of 18,324,206 raw reads were obtained. After excluding external targets and removing chimeras, 13,587,085 16S rDNA V3-V4 reads were retained. Coverage was above 97% for all groups, indicating that our sparse sequencing depth was sufficient to assess the diversity of vaginal samples in the study (Appendix A).

### 3.2. Classification of ASV

The Venn diagram showed that 2618 ASVs defined the core ASVs, while the number of ASVs specific to the G1 group was much higher than those of the other groups, and after G1, the number of ASVs observed decreased gradually with parity (Figure 1).

### 3.3. Community Structure of Vaginal Microorganisms and Changes in Correlation with Fecundity

In Simmental cow samples from different parities, the vaginal core microbiome was dominated by phyla Firmicutes and Bacteroides, which were followed by *Actinobacteria* and *Proteobacteria* (Appendix A). We selected *Lactobacillus*, *Pseudomonas*, *Fusobacterium* and *Corynebacterium*, which were present at >1% relative abundance, for investigating specific changes of these microorganisms with parity. The relative abundance of *Lactobacillus* was highest at stage G1, gradually decreased during the subsequent increase in parity, and then began to pick up gradually in G3. The relative abundance of Gram-negative bacteria *Pseudomonas* increased with the increase in parity and reached the peak at the G2 stage, and then, it gradually decreased. The relative abundance of *Fusobacterium* varied from high to low, and it showed the highest relative abundance at the G6 stage. The relative abundance of *Corynebacterium* increased with the increase in parity, and this trend was different from other microbiota. In general, the vaginal microbiota of low-parity cows showed greater variation than that of high-parity cows, and this difference showed different trends at different stages. Specifically, the vaginal microbiota changed more in the low-parity cows, while in the other stages, the vaginal microbiota changed relatively little (Figure 2).

### 3.4. Alpha Diversity Analysis

The Chao1 index differed significantly between parity at G0, G1, G4 and G5 (Table 1). From G0 to G1 it decreased significantly, increasing to G5 and falling at G6, indicating that the number of postpartum vaginal species increased with parity, and the number of high-parity vaginal bacteria was richer. Pielou [23], Shannon, and Simpson only showed significant differences between G0 and G1 (Table 1). Prenatal and postnatal vaginal microbiota changed significantly, and the abundance and diversity of postpartum vaginal microorganisms did not differ significantly.

### 3.5. Beta Polymorphism Analysis

Primary coordinate analysis (PCoA) shows that in the weighted case, the individual samples tend to be clustered but not tightly packed. In addition, the G1 group exhibited more pronounced dispersion. In the non-weighted case, the individual samples show greater dispersion. And each sample is distributed in different locations (Figure 3). 

### 3.6. LEfSe

Significant variations in species were only observed between the G0 group and higher parity groups. Osillospirales and UCG-010 were significantly more abundant in the G0 group, while only Acinetobacter showed a higher relative abundance in the G4 group. Methanobacteria exhibited a higher relative abundance in the G5 group, and Caviibacter appeared to be more prevalent in the G6 group compared to other groups (Figure 4).

### 3.7. Association between Functional Properties of the Vaginal Microbiome and Age

Functional prediction using PICRUST2 showed that G0 had higher carbohydrate metabolism and membrane transport (*p* < 0.05), while G1 showed higher amino acid metabolism (*p* < 0.05), comparing G0 with G1. Comparing G0 with G2, G2 showed stronger amino acid metabolism (*p* < 0.05), and G0 had higher nucleotide metabolism (*p* < 0.05), and comparing G0 with G3, G3 amino acid metabolism and lipid metabolism and glycan biosynthesis metabolism were higher (*p* < 0.05). Comparing G0 with G5, G0 had higher amino acid metabolism and folding classification and degradation of genetic information (*p* < 0.05). Comparing G0 with G6, the metabolism of the G0 enzyme family was higher (*p* < 0.05), while the endocrine was higher in G6 (*p* < 0.05). G1 was higher in nucleotide metabolism and folding sorting and the degradation of genetic information compared to G2 (*p* < 0.05), while G2 was higher in cell motility (*p* < 0.05). Carbohydrate metabolism and membrane transport were higher in G1 compared with G4 (*p* < 0.05). G1 was higher in carbohydrate metabolism and cell motility compared with G5 (*p* < 0.05). Comparing G1 with G6, G6 membrane transport was higher (*p* < 0.05). Comparing G2 with G4, G4 had stronger carbohydrate metabolism and genetic information processing, while G2 had higher glycan biosynthesis metabolism and signal transduction (*p* < 0.05). G2 was higher in amino acid metabolism and lipid metabolism compared to G5 (*p* < 0.05), while G5 was higher in genetic information processing and nucleotide metabolism (*p* < 0.05). G2 was higher in genetic information processing, nucleotide metabolism and lipid metabolism when comparing to G6 (*p* < 0.05). Comparing G3 with G4, G3’s cellular processes and signal transduction showed higher glycan biosynthesis metabolism (*p* < 0.05). G3 was higher than G5 in genetic information transcription (*p* < 0.05), and G3 was also higher in lipid metabolism, glycan biosynthesis and amino acid metabolism (*p* < 0.05). G3 secondary metabolic biosynthesis was higher than that of G6 (*p* < 0.05). G5 was higher in endocrinology than G6 (*p* < 0.05) (Figure 5 and Figure 6). 

## 4. Discussion

This study compared vaginal microbial differences in cows of different parities and explored their potential association with vaginal health. The core microbiome of bovine vaginas consists of four phyla, Firmicutes and Bacteroides being the most dominant, which were followed by *Actinobacteria* and *Proteobacteria* [24]. *Firmicutes*, accounting for a relatively high relative abundance in all parities and being the core group of vaginal microbiota, plays an important role in maintaining vaginal microbial balance and acid–base balance [25].

Changes in the vaginal microbiome can affect the reproductive health of cows [26]. Some harmful strains of bacteria, such as *Streptococcus*, *E. coli*, and *P. pyogenea*, can affect the uterus and ovaries through infection, leading to inflammation and the development of infection. These diseases not only affect the reproductive health of cows but can also lead to fetal dysplasia and fetal death, thereby prolonging the gestational period [27]. In addition, an imbalance in the vaginal microbiome can affect progesterone levels and disrupt ovarian function and ovulation, potentially leading to a prolonged interval between pregnancy and reproductive complications [28].

Hormonal changes can also affect microbes in the vagina. In cows, fluctuations in progesterone levels can play a significant role in maintaining the stability and overall health of the vaginal microbiome [29]. These findings highlight the importance of studying longitudinally the variation in bovine vaginal microbiota [30].

Research has shown notable variations in the vaginal microbiome between cows that have given birth and those that have not [24,31]. Similarly, the cows in our study experienced similarly significant changes in their vaginal microbiome after their initial calving. This might be attributed to the process of microbiome development during the first birth of a cow as well as the significant changes in the cow organs, hormones, and metabolism that occur during this time [32,33].

In general, there is a relationship between cow parity and their lactation function [34]. In terms of production, it is roughly divided into three stages: youth (G0–G2), maturity (G3–G4) and recession (G5–G6). The expected result of this study is that vaginal bacteria in these three periods have differences in diversity. Although the results have a certain deviation from our expectation, we have observed that the total number of vaginal bacteria species is different in individual groups (G0, G1, G4, and G5). Overall, the total number of vaginal microbial species also increased over the course of parity, but the correlation between the two needs to be further confirmed. However, the observed number of ASV decreases with increasing parity, and a possible explanation is that in high-parity cows, the vaginal microbiota has reached a relatively stable state, retaining the vaginal microorganism that plays a major role in the body, while the vaginal microbiota of little significance to cows gradually decreases or disappears. As the number of births in cows increases, changes in the vaginal environment during aging may breed more bacterial with low abundance, resulting in higher births and higher microbial species in cows.

Of course, this does not exclude the reason for this result: namely that we did not take the best sampling method. After all, it is very difficult to track the same cow for 7–8 years. Cattle production is typically phased out within three years, and project timelines make it almost impossible to achieve a perfect sampling method, which is why there is such a lack of longitudinal studies of cows’ vaginal microbes. Our research has tried to harmonize the living environment and feeding conditions of these cows in the hope that it will provide a reference for longitudinal studies of vaginal bacteria in dairy cows to some extent.

In addition, for the sampling time point, we were able to sample at the same time period to reveal microbial changes between cows of different parities. If sampling is not carried out at the same time, it may cause the experimental results to have effects or changes that do not exist, and only the experimental results obtained by sampling at the same time are of great significance.

After the first calving, the number of vaginal microbiota increases substantially [5]. This increase is associated with changes in the microbiota present in the vagina and uterus of Simmental cows [5]. The vaginal microbiome of first-time calving cows is more abundant in lactobacilli and has a high production of lactic acid, creating an ideal environment for their growth of the endometrium [35]. This also suggests that first-time calving may be a key point in regulating cow production and microbial stability. However, ASV in the vaginal microbiota decreases with increasing parity, which may be related to age, physical condition, and hormonal changes [36]. After the first calving, when the cow physical condition has not yet declined, the vaginal microbiota remains relatively stable. However, the number of vaginal microbes gradually decreases as the number of parities increases and the function of the ovaries and other reproductive organs decreases [30]. In addition, greater variation of the microbiota was found in G1–G3, and based on the results of this study, we speculate that the vaginal microbiota is less stable in low parity and more stable in high parity, but this still needs further validation.

In general, changes in the microbiota at the first calving are acceptable because the mother changes greatly due to calving at this time, but after that, the microbiota changes in lower parity are still not very stable and are greater. This suggests that changes in the vaginal microbiota due to childbirth may take longer to stabilize. Difference analysis showed that the composition of the microbiota of all groups was similar, but the abundance of individual microbiota varied with the number of parities. Lactobacilli in the vagina of first-calving cows did not show higher relative abundance compared to previous studies of dairy cows [37], and the proportion of Gram-negative bacteria in multi-prolific cows was lower [5]. In this study, beneficial lactobacilli were the most abundant in G0, then decreased with increasing parity until G3, and then increased again with increasing parity. In addition, among the more abundant Pseudomonas, the change in Gram-negative Pseudomonas was the most pronounced, from G0 to G2 with the increase in parity, and the relative abundance continued to decrease with the increase in parity. 

Studies have shown a link between uterine disease and the vaginal microbiota [38], which is usually composed of a variety of bacteria to form a balanced state. This balance is important for maintaining vaginal health (REFERENCE). However, when inflammation occurs in the endometrium, it may cause changes in the vaginal microbiome. Endometrial inflammation can lead to an imbalance in the vaginal microbiome, increasing the number of pathogenic bacteria and decreasing the number of beneficial bacteria. This imbalance can further exacerbate endometrial inflammation, creating a vicious circle. In some inflammatory diseases, such as endometritis, links to cow vaginal microbes have been identified [38,39]. In this study, according to the changes in the microbiota, the abundance of Gram-positive bacteria was highest at the peak of lactation. These bacteria are often defined as one of the factors that trigger inflammation, and G2, G3, and G4 may have local inflammation that we have not identified. In addition, beneficial bacteria such as *Lactobacillus* did not show high relative abundance during the whole process, and their specific role in the maintenance of vaginal homeostasis in dairy cows is still unclear.

Functional analysis showed that amino acid metabolism was generally enhanced in all cows that had given birth at least once, and endocrine enhancement in high-parity cows might be due to decreased ovarian function after multiple pregnancies, resulting in elevated estrogen and progesterone levels, and endocrine enhancement. From first birth to the G6 stage, the functional abundance of the vaginal microbiota showed a downward trend, and this decline was inversely correlated with parity. This analysis suggests that this might be related to changes in microbial function during maternal adaptive regulation [36]. In addition, Picrust2 results showed that *Methanobacteria* and *Caviibacter* might be related to amino acid metabolism and endocrine function, but this still needs further research.

## 5. Conclusions

Changes in the vaginal microbiome are part of a dynamic process, and the comparison of vaginal microbiome differences in cows of different parities can help to better understand the trends of the vaginal microbiome and its relationship to reproductive health. In addition, the interactions and effects of the cow’s vaginal microbiome with other environmental factors, such as feeding patterns, feeding, and disease, require further study. All of these factors may influence the composition and dynamics of a cow’s vaginal microbiota, thereby affecting their reproductive health and birth spacing [6,17].

Some progress has been made in the study of cow vaginal microbes, but there are still problems such as the small amount of data and lack of standardization compared to microbiological studies in humans and other animals [40]. Therefore, future research directions include establishing a more complete database, developing more efficient and accurate sequencing technology, establishing standardized data analysis methods, and conducting more in-depth research on the impact of different factors on cow vaginal microbiota so as to provide a more scientific basis for ensuring cow reproductive health and agricultural production.

## Figures and Tables

**Figure 1 animals-13-02880-f001:**
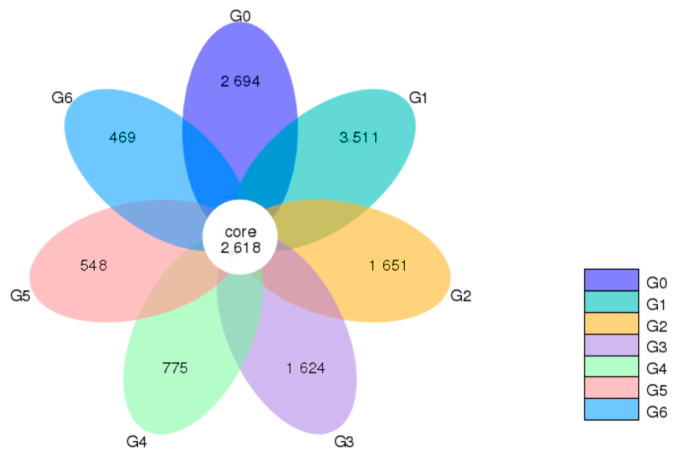
Venn plot plotted by ASV of cows of different parities. The “G” plus a numeric suffix indicates different parities. Based on the observed ASV, the data for each group were aggregated and plotted into a Venn plot to reflect the similarity and overlap of the microorganisms in each group.

**Figure 2 animals-13-02880-f002:**
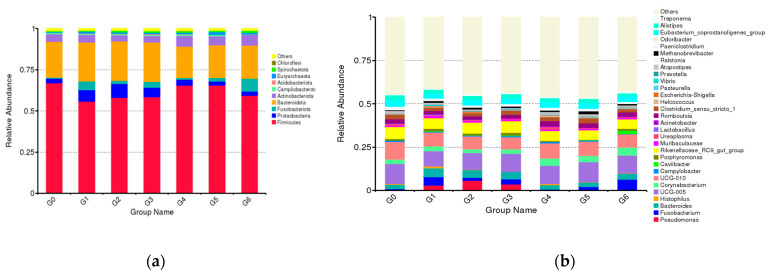
Relative abundance of each bacterial taxa at the phylum (**a**) and genus level (**b**) of cows at different parities. The “G” plus the numeric suffix represents different parities and the columns indicate the relative proportion of each parity.

**Figure 3 animals-13-02880-f003:**
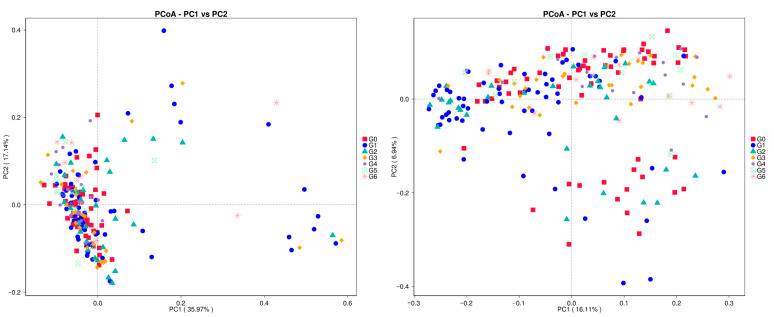
PCoA analysis of vaginal microbiota of dairy cows of different parities: based on the left-sided weighted Unifrac index and the unweighted Unifrac index on the right. Different types of points represent different groups, and the correspondence between specific points and groupings is on the right side of the figure.

**Figure 4 animals-13-02880-f004:**
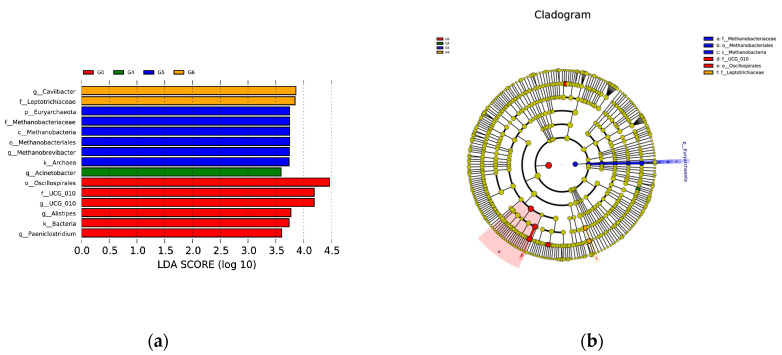
Microbial differences in bacterial communities between groups classified by parity size. (**a**) Bacterial classification of Simmental cow in different parities (**b**) Branching plots from linear discriminant analysis of effect sizes in seven groups.

**Figure 5 animals-13-02880-f005:**
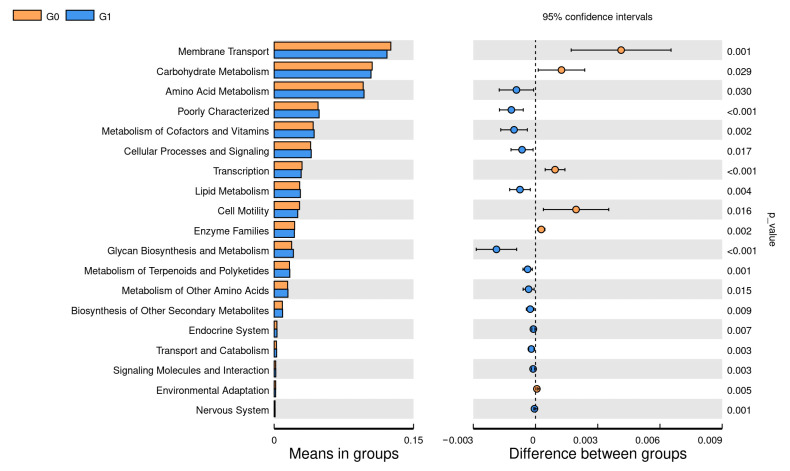
The pairwise comparison of the functional differences between each cycle obtained by PICRUST2, the expression of the different metabolic pathways on the left, and the corresponding *p*-value on the right; here is the functional predictive analysis comparison of G0 and G1 microbiota.

**Figure 6 animals-13-02880-f006:**
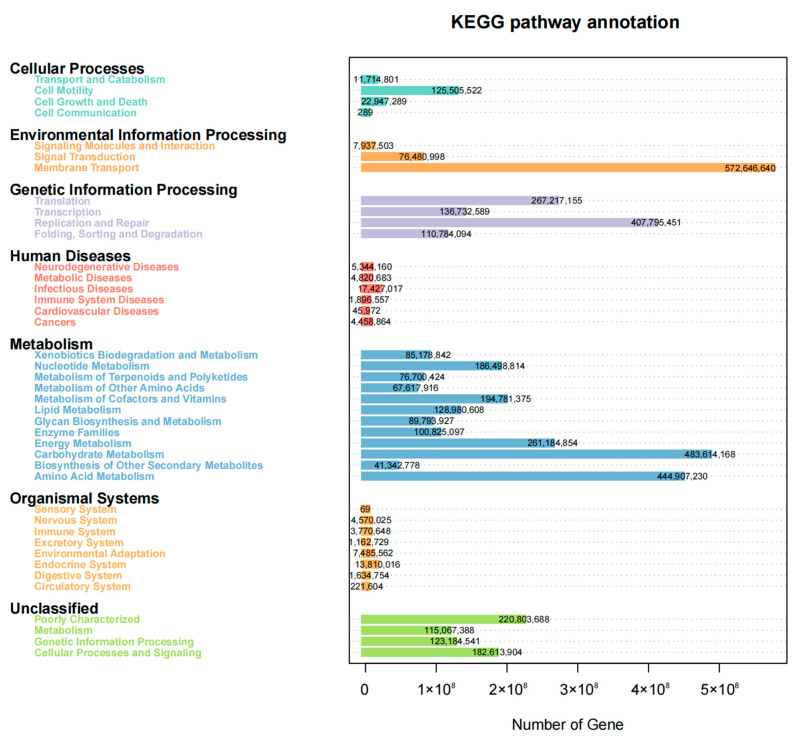
KEGG pathway annotation of all cow vaginal microbial samples with the metabolic pathway on the left and the number of genes expressed in the corresponding metabolic pathway on the right.

**Table 1 animals-13-02880-t001:** Average alpha diversity indexes in cows with different numbers of parities. A Tukey post hoc test was used to test differences between parities. “G” represents the cohort of the cow with similar parities.

Group	Shannon	Simpson	Chao1	Pielou
G0	9.31 ± 0.35 **	0.995 ± 0.004 *	1412.17 ± 174.69 **	0.89 ± 0.025 *
G1	8.40 ± 1.79 **	0.96 ± 0.08 *	1226.41 ± 345.40 **^ab^	0.82 ± 0.15 *
G2	8.75 ± 1.43	0.97 ± 0.06	1342.26 ± 302.81	0.84 ± 0.12
G3	8.75 ± 1.69	0.97 ± 0.08	1382.55 ± 328.73	0.84 ± 0.14
G4	9.24 ± 0.43	0.99 ± 0.004	1467.58 ± 219.76 ^b^	0.88 ± 0.03
G5	9.29 ± 0.56	0.99 ± 0.01	1537.48 ± 161.72 ^a^	0.88 ± 0.05
G6	8.79 ± 1.54	0.97 ± 0.065	1467.76 ± 269.76	0.84 ± 0.13

“**” indicates highly significant (*p* < 0.01), “*” indicates a significant difference (*p* < 0.05), in the same column “^a^” indicates highly significant (*p* < 0.01), “^b^” indicates a significant difference (*p* < 0.05).

## Data Availability

All the figures and tables used to support the results of this study are included.

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
