# Peer review of "Vaginal Microbiome Dynamics of Cows in Different Parities"

_animals, 2023, doi:10.3390/ani13182880_

Round 1
Reviewer 1 Report (New Reviewer)
The presented paper deals with the dynamic changes of vaginal microbiome during service period in milk cows of different parities. The Authors, using 16S rRNA sequencing technology, evaluated variability of bacterial community in samples obtained from calving cows of different number of births (from 1 to 6), whereas heifers 10-12 months old before parturition served as a control. This topic is current and interesting and, of course, is within the scope of the journal. The study attempts to make an advance to the better understanding of vaginal microbiome role for the reproductive health. The obtained results have a great potential as a start point for the future studies. The Author’s observed significant changes in vaginal microbes after first calving, as well as a gradual decrease, but at the same time a growing stability, number of the functional vaginal bacteria with increasing parity.
The general study design, scientific assumption and methodology are satisfactory. The entire manuscript is well-structured and relatively easy to follow, however I have some concerns regarding its editorial form and writing style:
- Discussion – this chapter is overlong and sometimes repeats the same aspects. Please, try to avoid the overinterpretation of your results, because you studied only one factor (vaginal bacteria) of reproductive health, which is a complex phenomenon. In my opinion, there is too much speculation in your statements. You should also consider bovine endometritis as a main potential negative consequence of vaginal bacteria, as well as role of uterine and vaginal self-healing mechanisms, that are very active beyond parturition,
- Material and methods (animal and sample collection) – please, give some details regarding the reproductive history of the experimental cows – e.g. course of postpartum, ovarian cyclicity, insemination etc. These factors are a potential causes of the vaginal bacteria variability. You wrote that the steroid hormones may influence the vaginal microbiome. The question is, were your cows in the same cycle phase (follicular or luteal) during sampling?,
- in my opinion, manuscript as a whole would benefit from an additional language check. Some phrases are not precise and difficult to understand.
In conclusion, because of its scientific novelty and a few other advantages, this paper is acceptable for publication, however, there is still room for its improvement. In my opinion some revisions and rewriting (mainly discussion) are required as a major revision.
In my opinion, English should be improved.
Author Response
First of all, thank you for your valuable comments and positive comments on this article,
- Discussion – this chapter is overlong and sometimes repeats the same aspects. Please, try to avoid the overinterpretation of your results, because you studied only one factor (vaginal bacteria) of reproductive health, which is a complex phenomenon. In my opinion, there is too much speculation in your statements. You should also consider bovine endometritis as a main potential negative consequence of vaginal bacteria, as well as role of uterine and vaginal self-healing mechanisms, that are very active beyond parturition,
Response:For the discussion part, it has been modified according to your comments, the duplicate and unwanted parts have been deleted, some of the speculations in the discussion are assumptions that integrate the overall situation to the causes of the results, and as you said in this study, we do not have the means to identify these ideas, so these conjectures are only placed as conjectures in the discussion section, and in this revision, we have also considered the over-interpretation of the parts and deleted some of them, while maintaining completeness while keeping the language as simple as possible.
For endometritis or other diseases and the impact on vaginal health, this direction is an important direction for studying the vaginal microorganisms of dairy cows, but there is no specific means to clarify whether they occur in this study, so it is not described in the article.
- Material and methods (animal and sample collection) – please, give some details regarding the reproductive history of the experimental cows – e.g. course of postpartum, ovarian cyclicity, insemination etc. These factors are a potential causes of the vaginal bacteria variability. You wrote that the steroid hormones may influence the vaginal microbiome. The question is, were your cows in the same cycle phase (follicular or luteal) during sampling?,
Response:Some of the details of your reproductive history have been added to the article, and in addition, the effect of hormones on vaginal microbes is a proven result, and in our study, we have tried to ensure that cows are in the same cycle by limiting sampling time.
- in my opinion, manuscript as a whole would benefit from an additional language check. Some phrases are not precise and difficult to understand.
Response:For the language aspect, we did a language check again in the hope of improving the overall quality of the article.
Reviewer 2 Report (New Reviewer)
I carefully studied the manuscript. As a general organization it is correct. But there are some things to redo and others are in confusion.
- to bring clearer information to the groups of cows with criteria for inclusion and exclusion from the lot depending on the condition of the cervix
- the comments must be rewritten, confusion with uterine flora/infection and the effects on fertility.
attached in yellow, more comments are highlighted.
overview: in the manuscript, the following are necessary: graphic corrections, technical editing, details of abbreviations.
The author confuses the vaginal flora with the uterine flora, especially in the discussion chapter. There are distinct issues in the puerperium (60-90 d), the vaginal flora may differ from the uterine flora, the condition is essential and related to the condition of the cervix.
There are also confusions in citing the bibliography - vaginal flora vs uterine flora.
I followed the bibliography and cited references, the vast majority of them refer to contamination and uterine flora. The author uses it wrongly as relevant to vaginal flora / contamination / vaginal conditions.
L11: UNCLEAR. I find this term a bit pretentious. When I read unclearly, it makes me think of something little studied without knowing things with certainty. In my opinion, things are not like that. Please find another synonym.
L 23-24, I do not agree with this expression
L 25 Detail what you are referring to. You can add additional information to the introductory section, after lines 72-75.
L29-30-31-34: what's the difference. Numbers are also used in the abstract.
L71-72: Additional information related to crross-section/vertical and longitudinal studies should be provided. The reader must understand what he is referring to. Definitions and examples would be necessary.
L77 to explain the term -longitudinal time and space
L 82-90. For category G 1-6, depending on parity, additions must be made about the evolution of the puerperium, if in the interval up to 60-90 days they had RP, Endometritis, Lochiometritis, uterine subinvolution, any deviation from the physiological puerperium
What are the conditions of accommodation, food and the maintenance regime, do they have free access to movement space.....
Also BCS's physical condition and general state of health. Or the presence of other comorbidities.
At the time of collection, were there cows with vulva-vaginal secretions? have you included / excluded them in groups?
L 147. I do NOT see any numbers on this graph. Graphic too small, rebuild
L 226: correct it. Ascending infections do not reach the ovaries, only the inflammatory process can reach the periovarian, not the ovarian infection. The ovary does not have a tubular structure like the salpinx and uterine horns. Cases of microbial (bacterial/fungal) ovariitis are very rare in cows, especially in your cows included in groups G1-6.
L 269-270: Bring information and examples of how reproductive hormonal status is influenced by vaginal flora!

Author Response
I carefully studied the manuscript. As a general organization it is correct. But there are some things to redo and others are in confusion.
- to bring clearer information to the groups of cows with criteria for inclusion and exclusion from the lot depending on the condition of the cervix
Response: First of all, thank you for your valuable comments and constructive questions for this article, according to your comments we have added more information about the dairy herd and, in addition, for the condition of the cervix, We don't have a very strict definition of a healthy cow, but at least we do guarantee that the cow is able to produce normally and without obvious pathological physiology that affects daily life.
- the comments must be rewritten, confusion with uterine flora/infection and the effects on fertility.
attached in yellow, more comments are highlighted.
Response:I don't quite understand what you mean by "comments"
overview: in the manuscript, the following are necessary: graphic corrections, technical editing, details of abbreviations.
The author confuses the vaginal flora with the uterine flora, especially in the discussion chapter. There are distinct issues in the puerperium (60-90 d), the vaginal flora may differ from the uterine flora, the condition is essential and related to the condition of the cervix.
There are also confusions in citing the bibliography - vaginal flora vs uterine flora.
I followed the bibliography and cited references, the vast majority of them refer to contamination and uterine flora. The author uses it wrongly as relevant to vaginal flora / contamination / vaginal conditions.
Response:For the references in this article, when looking for literature, there are very few studies on longitudinal studies of cow vaginal microorganisms, and most of the literature on cow reproduction is also about the uterus, which is why the number of vaginal microorganism literature in this article is small, in addition, the flora in the discussion section of this article is not accounted for as uterine flora and only defines it as reproductive tract flora or vaginal flora, and the specific content of this article cannot be defined because of the cited literature.
L11: UNCLEAR. I find this term a bit pretentious. When I read unclearly, it makes me think of something little studied without knowing things with certainty. In my opinion, things are not like that. Please find another synonym.
Response:Modified based on your comments
L 23-24, I do not agree with this expression
Response:Modified based on your comments
L 25 Detail what you are referring to. You can add additional information to the introductory section, after lines 72-75.
L29-30-31-34: what's the difference. Numbers are also used in the abstract.
Response:What is described here is that the observed number of vaginal microbial ASVs decreases with increasing parity, while the number of species increases.
L71-72: Additional information related to crross-section/vertical and longitudinal studies should be provided. The reader must understand what he is referring to. Definitions and examples would be necessary.
L77 to explain the term -longitudinal time and space
Response:The definition of longitudinal spatiotemporal here refers to the comparison of physiological conditions in different periods of time with time variables, and in this case refers to different parity.
L 82-90. For category G 1-6, depending on parity, additions must be made about the evolution of the puerperium, if in the interval up to 60-90 days they had RP, Endometritis, Lochiometritis, uterine subinvolution, any deviation from the physiological puerperium
What are the conditions of accommodation, food and the maintenance regime, do they have free access to movement space.....
Also BCS's physical condition and general state of health. Or the presence of other comorbidities.
At the time of collection, were there cows with vulva-vaginal secretions? have you included / excluded them in groups?
Response:Based on your comments, we have supplemented the specific information of the puerperium, and for vulvar secretions, we clean the cow's vulva before collection and use a longer cotton swab for sampling.
L 147. I do NOT see any numbers on this graph. Graphic too small, rebuild
Response:Modified based on your comments.
L 226: correct it. Ascending infections do not reach the ovaries, only the inflammatory process can reach the periovarian, not the ovarian infection. The ovary does not have a tubular structure like the salpinx and uterine horns. Cases of microbial (bacterial/fungal) ovariitis are very rare in cows, especially in your cows included in groups G1-6.
Response:Modified based on your comments.
L 269-270: Bring information and examples of how reproductive hormonal status is influenced by vaginal flora!
Response:Changes in vaginal microbes can cause hormonal changes that affect the ovarian function of cows, and cow hormone levels also affect changes in microbial abundance, for example, the presence of vaginal microorganisms can affect the pH of the vagina. Normally, the pH of the cow's vagina should be kept in a low range (about 5.5 to 6.5). When the vaginal microbiota is out of balance, it may lead to an increase in vaginal pH, which in turn affects ovarian function and ovulation, and certain microorganisms in the vagina may produce metabolites such as Volatile fatty acids (VFAs), which may have an impact on ovarian function and ovulation in dairy cows. Some research suggests that certain VFAs may affect ovulation in cows by regulating the secretion of ovarian hormones. It is important to note that the relationship between cow vaginal microbes and ovulation in the body needs further research to be fully understood.
Reviewer 3 Report (New Reviewer)
This study demonstrated the changes in vaginal microbiome according to the parity in cows. It was found that the variation of vaginal flora in high-calving cows was less than that in low-calving cows.
I have the following concerns.
Major Comments
1. Please indicate that the approval of the animal experiment plan has been obtained.
2.Sampling was noted in the discussion section, feeding was uniform and during the same period.
In the method, please indicate that the sampling period is the same period. Also, please add information about the cattle rearing situation, such as whether it is free stall or tie stall, separate feed or TMR.
3.You mentioned that you used healthy cows, but what specific diseases are excluded? For example, are you testing for endometritis? Also, can it be collected from the vagina without touching the vestibule of vagina?
4.Sampling is performed from the vagina, not from the vestibule of vagina. How about showing the length of the cotton ball shaft?
5. As the author also mentions in the discussion, the vaginal microbiome may change depending on whether it is the estrous period or the luteal period.
Regarding this sampling, have you checked the estrous cycle or the presence or absence of the corpus luteum?
Minor Coments
1. Line 177 between parands
I didn't understand what "parands" meant.
2.Table 1
What is the reason for not aligning the decimals?
Author Response
First of all, thank you for your valuable feedback,
- Please indicate that the approval of the animal experiment plan has been obtained.
2.Sampling was noted in the discussion section, feeding was uniform and during the same period.
In the method, please indicate that the sampling period is the same period. Also, please add information about the cattle rearing situation, such as whether it is free stall or tie stall, separate feed or TMR.
Response:It has been modified and supplemented based on your comments。
3.You mentioned that you used healthy cows, but what specific diseases are excluded? For example, are you testing for endometritis? Also, can it be collected from the vagina without touching the vestibule of vagina?
Response:We don't have a very strict definition of a healthy cow, but at least we do guarantee that the cow is able to produce normally and without obvious pathological physiology that affects daily life.
4.Sampling is performed from the vagina, not from the vestibule of vagina. How about showing the length of the cotton ball shaft?
Response:The sampling is carried out inside the vagina, modified in the method, the length of the cotton swab is about 30 cm.
- As the author also mentions in the discussion, the vaginal microbiome may change depending on whether it is the estrous period or the luteal period.
Regarding this sampling, have you checked the estrous cycle or the presence or absence of the corpus luteum?
Response:In this study, no specific estrus cycle or the presence of corpus luteum were investigated, but we have described in the methodology the specific sampling time of all cows to ensure that they are in the same cycle as much as possible.
Minor Coments
- Line 177 between parands
I didn't understand what "parands" meant.
Response:There may have been some omissions in the writing process, which have been modified.
2.Table 1
What is the reason for not aligning the decimals?
Response:The reason for the lack of alignment may be that some data is superscript and others are not.
Round 2
Reviewer 1 Report (New Reviewer)
In my second review of this paper, I have to say, that overall evaluated manuscript has been improved. A few items raised by reviewers before have been addressed and it makes the paper easier to follow.
In general, I still stand by my earlier opinion, that this paper is worthy of being published in the journal. However, in my opinion, there is still room for its improvement. Currently, during the proces of evaluation, we are working together with Authors how to improve the editorial form of this paper. I have the impression that the Authors did not fully understand my earlier remark regarding endometritis. Because, in your earlier discussion you speculated a lot about the relationship between vaginal microbiome as well as a general animal health and productivity (weakly related to main topic), I suggested to consider above all the clinical and/or subclinical endometritis as a immediate consequence of the vaginal microbiome changes. A close relationships between these two factors are well known. Additionally, taking into account a high incidence of this uterine disorder in milk cows herds and its very high potential to influence vaginal microbiome, in my opinion, this factor is discussion worthy. Moreover, you should also consider this high prevalence of uterine infections (clinical or subclinical) in a description of a whole reproductive history of your experimental animals. It is unbelievable that during the sampling period lasting over 6 years in so numerous cows, this disorder did not occur as a potential factor influencing vaginal microbiome. However, I am aware, that you are not able to change this situation and your study design, but in my opinion, you should consider this aspect as a disadvantage and/or limitation of your study.
In general, I recommend this paper for publication with minor revision, however, the final decision should rest with Editor.
Author Response
In my second review of this paper, I have to say, that overall evaluated manuscript has been improved. A few items raised by reviewers before have been addressed and it makes the paper easier to follow.
In general, I still stand by my earlier opinion, that this paper is worthy of being published in the journal. However, in my opinion, there is still room for its improvement. Currently, during the proces of evaluation, we are working together with Authors how to improve the editorial form of this paper. I have the impression that the Authors did not fully understand my earlier remark regarding endometritis. Because, in your earlier discussion you speculated a lot about the relationship between vaginal microbiome as well as a general animal health and productivity (weakly related to main topic), I suggested to consider above all the clinical and/or subclinical endometritis as a immediate consequence of the vaginal microbiome changes. A close relationships between these two factors are well known. Additionally, taking into account a high incidence of this uterine disorder in milk cows herds and its very high potential to influence vaginal microbiome, in my opinion, this factor is discussion worthy. Moreover, you should also consider this high prevalence of uterine infections (clinical or subclinical) in a description of a whole reproductive history of your experimental animals. It is unbelievable that during the sampling period lasting over 6 years in so numerous cows, this disorder did not occur as a potential factor influencing vaginal microbiome. However, I am aware, that you are not able to change this situation and your study design, but in my opinion, you should consider this aspect as a disadvantage and/or limitation of your study.
In general, I recommend this paper for publication with minor revision, however, the final decision should rest with Editor.
Response:First of all, thank you for your positive comments on this article and your support of the review, as you said, there is indeed a lack of discussion about disease in this article, and after reviewing the literature, we have added some discussion about uterine disease and inflammation to the discussion section in the hope of better describing the vaginal microbes of dairy cows and their relationship to cow health. In addition, we have tried our best to control the details of the breeding history of dairy cows due to various restrictions, which can only be controlled to a certain extent.
Finally, thank you again for your constructive comments on this article and for guiding the revision work.
Reviewer 2 Report (New Reviewer)
I have read the contributions made by the author, they have made changes but there are still limitations. I stick to my first opinion.
The authors did not respond to every comment / remark. For some remarks, there are only answers in the letter but not the changes in the manuscripts.
I recommend to the authors, the resumption of the comments from R1.
In the text, when discussing vaginal contamination, we must cite an article with vaginal flora, not one with uterine flora.
Author Response
First of all, thank you for your constructive comment on this article, we once again reply to your comments and supplement, in addition, the reason why some of the comments are incomplete is because the responses of some comments together may be easier to understand and have a holistic nature, but here we once again add the replies to some comments, the specific content is as follows: I carefully studied the manuscript. As a general organization it is correct. But there are some things to redo and others are in confusion.
- to bring clearer information to the groups of cows with criteria for inclusion and exclusion from the lot depending on the condition of the cervix
Response: First of all, thank you for your valuable comments and constructive questions for this article, according to your comments we have added more information about the dairy herd and, in addition, for the condition of the cervix, We don't have a very strict definition of a healthy cow, but at least we do guarantee that the cow is able to produce normally and without obvious pathological physiology that affects daily life.
- the comments must be rewritten, confusion with uterine flora/infection and the effects on fertility.
Response:After reviewing the literature, we added information about uterine flora/infection confusion and the impact on fertility in the discussion section.
attached in yellow, more comments are highlighted.
Response:Modified based on your comments
overview: in the manuscript, the following are necessary: graphic corrections, technical editing, details of abbreviations.
Response:Modified based on your comments
The author confuses the vaginal flora with the uterine flora, especially in the discussion chapter. There are distinct issues in the puerperium (60-90 d), the vaginal flora may differ from the uterine flora, the condition is essential and related to the condition of the cervix.
Response:The flora discussed in this article does not count as uterine flora, but is defined only as genital tract flora or vaginal flora, and the specific content of this article should not be defined by the cited literature.
There are also confusions in citing the bibliography - vaginal flora vs uterine flora.
I followed the bibliography and cited references, the vast majority of them refer to contamination and uterine flora. The author uses it wrongly as relevant to vaginal flora / contamination / vaginal conditions.
Response:For the references in this article, when looking for literature, there are very few studies on longitudinal studies of cow vaginal microorganisms, and most of the literature on cow reproduction is also about the uterus, which is why the number of vaginal microorganism literature in this article is small, in addition, the flora in the discussion section of this article is not accounted for as uterine flora and only defines it as reproductive tract flora or vaginal flora, and the specific content of this article cannot be defined because of the cited literature.
L11: UNCLEAR. I find this term a bit pretentious. When I read unclearly, it makes me think of something little studied without knowing things with certainty. In my opinion, things are not like that. Please find another synonym.
Response:Modified based on your comments
L 23-24, I do not agree with this expression
Response:Modified based on your comments
L 25 Detail what you are referring to. You can add additional information to the introductory section, after lines 72-75.
Response:What is described here is that the observed number of vaginal microbial ASVs decreases with increasing parity, while the number of species increases.
L29-30-31-34: what's the difference. Numbers are also used in the abstract.
Response:Modified based on your comments
L71-72: Additional information related to crross-section/vertical and longitudinal studies should be provided. The reader must understand what he is referring to. Definitions and examples would be necessary.
Response:In layman's terms, longitudinal space-time refers to the comparison of physiological conditions and time variables in different periods, combined with the context of this article, here refers to different parity.
L77 to explain the term -longitudinal time and space
Response:The definition of longitudinal spatiotemporal here refers to the comparison of physiological conditions in different periods of time with time variables, and in this case refers to different parity.
L 82-90. For category G 1-6, depending on parity, additions must be made about the evolution of the puerperium, if in the interval up to 60-90 days they had RP, Endometritis, Lochiometritis, uterine subinvolution, any deviation from the physiological puerperium
Response:Based on your opinion, we have supplemented the specific information of the puerperium.
What are the conditions of accommodation, food and the maintenance regime, do they have free access to movement space.....
Response:In the method section, we have added these details of what you said.
Also BCS's physical condition and general state of health. Or the presence of other comorbidities.
Response:When it comes to diseases, our minimum standards guarantee that cows are able to produce normally, without obvious pathological or physiological features that affect daily life.
At the time of collection, were there cows with vulva-vaginal secretions? have you included / excluded them in groups?
Response:Based on your comments, we have supplemented the specific information of the puerperium, and for vulvar secretions, we clean the cow's vulva before collection and use a longer cotton swab for sampling.
L 147. I do NOT see any numbers on this graph. Graphic too small, rebuild
Response:Modified based on your comments.
L 226: correct it. Ascending infections do not reach the ovaries, only the inflammatory process can reach the periovarian, not the ovarian infection. The ovary does not have a tubular structure like the salpinx and uterine horns. Cases of microbial (bacterial/fungal) ovariitis are very rare in cows, especially in your cows included in groups G1-6.
Response:Modified based on your comments.
L 269-270: Bring information and examples of how reproductive hormonal status is influenced by vaginal flora!
Response:Changes in vaginal microbes can cause hormonal changes that affect the ovarian function of cows, and cow hormone levels also affect changes in microbial abundance, for example, the presence of vaginal microorganisms can affect the pH of the vagina. Normally, the pH of the cow's vagina should be kept in a low range (about 5.5 to 6.5). When the vaginal microbiota is out of balance, it may lead to an increase in vaginal pH, which in turn affects ovarian function and ovulation, and certain microorganisms in the vagina may produce metabolites such as Volatile fatty acids (VFAs), which may have an impact on ovarian function and ovulation in dairy cows. Some research suggests that certain VFAs may affect ovulation in cows by regulating the secretion of ovarian hormones. It is important to note that the relationship between cow vaginal microbes and ovulation in the body needs further research to be fully understood.
Response : Here again to reply to the question about the literature, there will also be discussions about vaginal flora in the literature related to endometrial inflammation, so the author believes that the cited literature does not have to be completely related to vaginal microbes.
This manuscript is a resubmission of an earlier submission. The following is a list of the peer review reports and author responses from that submission.
Round 1
Reviewer 1 Report
This manuscript provides novel information about how parity number influences the vaginal microbiome. Authors provide a good argument for why knowledge about the vaginal microbiome will assist with describing problems that may occur in the uterus. A few minor issues with English writing exist, but overall, this manuscript is nicely written. Some clarifications of methods are needed. These have been listed below.
The major concern with this work is that it appears that no efforts were made to take vaginal samples from cattle that the same (or similar) stages of lactation, pregnancy status, and/or stage of the estrous cycle. This lack of timing is potentially highly problematic. Ideally, all sampling would occur in cows that are at similar days post-partum (e.g., 70-100 days), all cows would be cycling or at a similar stage of pregnancy. Moreso, given that progesterone may influence the microbiome, sampling at the same stage of the estrous cycle is ideal if cyclic cows were used.
The suggestion is that authors include all the specific parameters they used for days post-partum, pregnancy status, and stage of the estrous cycle in the methods (Ln 78-82). Authors also should provide a comprehensive discussion about the potential pitfalls that exist if one or more of these parameters was not considered. This discussion point is very important. It will determine whether readers will view this work as having value or being fatally flawed.
Specific comments:
Ln 2: Title. If there is no special reason why Simmentals were used, I suggest you remove the breed name in the title. Perhaps say “…beef cows…” .
Ln 24: “it has been observed” and related phrases are not needed. Consider removing these.
Ln 27: “did not litter” is a term I am not familiar with. Did not give birth?
Ln 79: Why not use “P” instead of “G” to indicate parity number?
Ln 81-82: Provide more detail on vaginal sampling with special attention paid to how cross-contamination with the exterior of the vulva was ensured. Was a vulva sample taken to show that the microbial community being samples were not vaginal in origin?
Ln 85: It’s not clear how one can quantify DNA quality and concentration by agarose gel electrophoresis, or at least other than if you used a Bioanalyzer. Please explain this better.
Suggest you focus on making sure sentences are as concise as possible.
Reviewer 2 Report
The paper entitled "Vaginal microbiome dynamics of Simmental cow at different parities" evaluated the vaginal microbiome of Simmental cows. The paper states that most of the research on vaginal microbiome in cows were cross-sectional studies, and longitudinal studies were missing.
However, their study design was cross-over and not longitudinal. A longitudinal study evaluates the same subject over different time points. Using different cows at different parities does not characterizes a longitudinal study.
In addition, the methodology is poor described. Does a power calculation was done before the study started?
The number of animals in each parity group differs, and it could be a bias for the results of this study,
The methodology does not describe the cows, if they were lactating or not, how long after the last calving, season... Details that are very important for the interpretation of the results.
The introduction could be better written to describe the reasoning behind their study. A review of the study design should be made, and the introduction needs to be changed accordingly. The methodology needs to be greatly improved.
Unfortunately, the paper can not be accepted in its current version.
Reviewer 3 Report
This study aims to investigate longitudinal changes in vaginal microbiome of Simmental cows according to parity. This is a novel concept and will provide important information for the field of bovine reproductive microbiome. There are however, a few aspects of this manuscript that must be changed for clarity, as well as some inconsistencies in the results that must be addressed. Please find my comments below.
Simple Summary
Lines 15-17: The way this is written is confusing. Was there no significant differences for all of those analyses?
Line 18: Do you mean Methanobacteria or methanosis?
Abstract
Line 25-26: Do you mean thath abundance decreases while the number of species increase (more diverse?)
Line 27: Did not calve rather than did not litter
Introduction
Lines 41-43: This is true for women, but not for livestock species. Please specify women instead of the general term females
Line 44: To my knowledge, Lactobacillus are in very low abundance in the vagina of cows.
Lines 48 - 49: The vaginal pH of cows is neutral to slightly alkaline (reference: https://doi.org/10.3389/fmicb.2016.01936 )
Materials and Methods:
Please clarify whether the cows in G6 are the same individuals as the ones in G0. i.e. this study was a long period study that sampled the same individuals over a period of 7 years, rather than identifying the number of births the animals had and sampling them. In addition, please include the years in which the study was performed and data were collected. Finally, please specify at which point the samples were collected (during pregnancy, immediately after calving, X many days after calving)
Lines 81-82: I suggest: "Samples were immediately placed on ice and hauled into the laboratory in the same day, where samples were stored at -80 until DNA was extracted"
Results
Some of your results are missing a P-values, such as in your beta diversity analyses, and your 3.2, and 3.3 sessions.
Line 139: I suggest: with a negative correlation with parity
Figure 1: What do you mean across fetuses? Do you mean across parities?
Line 144: parities, instead of paritys
Lines 144-146: The phylum with the greatest relative abundance, according to figure 2, was Firmicutes. Was that not present among all samples and was thus not classified as core microbiome? Please clarify inconsistencies between this text and your figure.
Lines 146-149: Why did you choose these bacteria in specific? Were they the most abundant in most samples? Were they the most abundant among your core microbiome?
Line 155-156: You state "the vaginal microbiota of low-parity cows showed greater variation than that of high parity cows" In which way did they show greater variation? In relative abundance? In diversity? In number of microorganisms?
Lines 157-158: You state "the vaginal microbiota changed more in the low-parity cows, while in the other stages, the vaginal microbiota changed relatively little: In which way did the microbiota change?
Lines 162 and 174: Why polymorphism analysis instead of diversity analysis?
Line 201: According to figure 4, Archea showed a higher relative abundance in the G5 group, not G4.
Lines 211 - 224: I suggest saying when comparing GX with GY, GX had stronger..., instead of GX compared with GY, GX had stronger...
Figure 5: You need a statement in the text referring to this figure.
Discussion
Line 247: How did you assess vaginal health?
Lines 247 - 251: I'm confused. You say that Firmicutes phylum is the core group and occupies relative high proportion, and according to figure 2, it is the one with the greatest abundance. Yet, on lines 247 - 249 you state that Thickobacterium and Bacteroidetes were the most dominant, yet these phyla are not even included in figure 2. Please clarify
Line 250: reword fetuses
Lines 261 - 263: Please include the source of that statement
Lines 272 - 273: "Differences" is repeated
Lines 282 - 283: Please include the reference for the statement "The vaginal microbiome of navial cows is more abundant in lactobacilli and has a stronger ability to produce lactic acid".
Line 290: "the vaginal microbiota shows a gradual decrease" in which way? in abundance of species? in diversity? in total number of microorganisms?
Lines 299-301: Is this in relation to a different study or to the present manuscript? If from different study, please include reference. If from the present study, what cycle are you referring to? estrous cycle? parity stage?
Lines 301 - 308: What was the relative abundance of lactobacillus and Pseudomonas in your study? In figure 2, the genus lactobacillus is not even shown in the figure
Lines 309 -310: You state that "Amino acid metabolism is generally enhanced in calving cows". What do you mean by calving cows? Cows of older parity? cows in the process of calving?
Lines 314 - 316: are you referring to the vaginal microbiome in women or in cattle? Please include the source
Conclusions
Lines 342 - 345. Sentence is confusing
Some sentences can be rewritten for clarity. There are often words written with capital letter when they should be written in lowercase (i.e., line 48: postpartum endometritis, Likewise,...) or (line 134: gene data analysis According, data...) and there are spaces before a dot (line 139: parity (Figure 1) . )
In addition, some sentences are repetitive and confusing (i.e. in lines 342 - 344 the word Simmental is used 4 times)
Finally, terms in relation to cattle should be used properly. Farrowing, and litter, for example, are not commonly used for cows, but rather calving and offspring. Throughout the text, some words are used either by mistake or out of context, such as "fetuses" and "fecundities"